# Post-Discharge Depression Status for Survivors of Extracorporeal Membrane Oxygenation (ECMO): Comparison of Veno-Venous ECMO and Veno-Arterial ECMO

**DOI:** 10.3390/ijerph19063333

**Published:** 2022-03-11

**Authors:** Wan-Jung Lin, Yu-Ling Chang, Li-Chueh Weng, Feng-Chun Tsai, Huei-Chiun Huang, Shu-Ling Yeh, Kang-Hua Chen

**Affiliations:** 1Department of Nursing, National Taiwan University Hospital, College of Medicine, National Taiwan University, Taipei 100, Taiwan; dyxz8k69022@gmail.com; 2School of Nursing, College of Medicine, Chang Gung University, Taoyuan City 333, Taiwan; ylchang@mail.cgu.edu.tw (Y.-L.C.); ax2488@mail.cgu.edu.tw (L.-C.W.); 3Department of Nursing, Chang Gung Memorial Hospital, Linkou Branch, Taoyuan City 333, Taiwan; 4Department of General Surgery, Chang Gung Memorial Hospital, Linkou Branch, Taoyuan City 333, Taiwan; 5Department of Cardiovascular Surgery, Chang Gung Memorial Hospital, Linkou Branch, Taoyuan City 333, Taiwan; lutony@cgmh.org.tw (F.-C.T.); hueichiun@cgmh.org.tw (H.-C.H.); 6College of Medicine, Chang Gung University, Taoyuan City 333, Taiwan; 7Department of Nursing, Chang Gung Memorial Hospital, Taoyuan Branch, Taoyuan City 333, Taiwan; q22122@cgmh.org.tw; 8Department of Nursing, Chang Gung University of Science and Technology, Taoyuan City 333, Taiwan

**Keywords:** cross-sectional design, extracorporeal membrane oxygenation, veno-arterial, veno-venous, cardiac failure, respiratory failure, depression status, mental health support

## Abstract

Extracorporeal membrane oxygenation (ECMO) is one of the common invasive treatments for the care of critically ill patients with heart failure, respiratory failure, or both. There are two modes of ECMO, namely, veno-venous (VV) and veno-arterial (VA), which have different indications, survival rates, and incidences of complications. This study’s aim was to examine whether depression status differed between patients who had received VV-ECMO or VA-ECMO and had been discharged from the hospital. This was a descriptive, cross-sectional, and correlational study of patients who had been discharged from the hospital at least one month after receiving ECMO at a medical center in northern Taiwan from June 2006 to June 2020 (*N* = 142). Participants were recruited via convenience and quota sampling. Data were collected in the cardiovascular outpatient department between October 2015–October 2016 (*n* = 52) and September 2019–August 2020 (*n* = 90). Participants completed the Hospital Anxiety and Depression Scale–Depression (HADS-D) as a measure of depression status. Post-discharge depression scores for patients who received VV-ECMO (*n* = 67) was significantly higher (*p* = 0.018) compared with participants who received VA-ECMO (*n* = 75). In addition, the mode of ECMO was a predictor of post-discharge depression (*p* = 0.008) for participants who received VV-ECMO. This study concluded that patients who received VV-ECMO may require greater mental health support. Healthcare professionals should establish a psychological clinical care pathway evaluated by multiple healthcare professionals.

## 1. Introduction

Extracorporeal membrane oxygenation (ECMO) is a common invasive treatment for respiratory failure, cardiac failure, or both [1], and is a cardiopulmonary life-supportive therapy rather than a disease-curing therapy. With the advancements of technology and the quality of care in the intensive care unit (ICU), the number of patients receiving ECMO has increased sharply in the past 10 years [2], as has the rate of survival.

There are two modes of ECMO, which differ according to medical indications. The Extracorporeal Life Support Organization (ESLO) recommends veno-arterial ECMO (VA-ECMO) for patients with refractory cardiogenic shock due to acute myocardial infarction, fulminant myocarditis, reactions to cardiotoxic drugs, end-stage dilated or ischemic cardiomyopathy, hypothermia with refractory cardiocirculatory instability, massive pulmonary embolism, or transplant acute cardiogenic shock [3]. During VA-ECMO, the ECMO circuit is connected in parallel to the heart and lungs, which allows blood to bypass both organs and is returned to the arterial system via peripheral cannulation [3,4]. Indicators for patients needing veno-venous ECMO (VV-ECMO) include severe, acute, reversible respiratory failure that is refractory to optimal medical management [5]. Venous cannulation is performed singly through the right internal jugular vein or with double venous cannulas placed in the common femoral vein and right internal jugular or femoral vein [4,5].

Both types of ECMO have their unique complications [1]. Bleeding complications were reported to be greater for patients who receive VA-ECMO (68.5%) compared with those who receive VV-ECMO (39.1%) [6], and the rate of hemolysis during therapy is significantly higher for patients receiving VA-ECMO (4%) versus 2% for VV-ECMO (*p* < 0.001) [7]. Outcomes also differ with the mode of ECMO for the rate of survival to hospital discharge, which is higher for VV-ECMO (63%) versus VA-ECMO (40%) [7]. Some of these differences in outcomes may be due to the indication for treatment. VA-ECMO is indicated for cardiac patients, who are at greater risk of death than patients with respiratory disease [8].

Psychological problems are common in patients with heart failure [9], acute respiratory distress syndrome [10], and critical illness [11]. Depression is one of the most common psychological problems, where it co-exists with chronic respiratory diseases [12], symptoms of dyspnea [13], acute respiratory distress syndrome (ARDS) [13,14], and disorders of anxiety and dementia [15]. Survivors of ARDS and patients with long periods of hospitalization have high levels of depression, anxiety, and post-traumatic stress syndrome for up to 5 years following hospital discharge [16,17], which may be associated with increased levels of mediators of inflammation [12], ventromedial prefrontal cortex activity [18], changes in components of the tryptophan–kynurenine metabolic response to inflammation [19], and brain hypoxia with neuronal abnormalities [20]. A study showed that neuronal function is highly sensitive to changes in oxygen levels, but how hypoxia affects dendritic spine formation and synaptogenesis. Because synaptic transmission is a high-energy-consuming process and neurons rely on oxygen to produce ATP, one possible mechanism whereby neurons can avoid an energy crisis is by decreasing synaptic transmission through spine regression [20].

Many patients experience constant depressive symptoms 1 year post-discharge following ECMO due to physical complications [21]. However, after discharge following ECMO, patients often experience poorer quality of life than the general population, delays in physical recovery, and a risk of psychological problems [21,22,23]. Many patients experience constant depressive symptoms 1 year post-discharge following ECMO due to physical complications [21]. Depression is a neurological illness with persistent symptoms, which can interfere with an individual’s quality of life following discharge from ECMO [22]. Symptoms can include sadness, a decrease in activities one used to enjoy, changes in weight, and difficulty sleeping, which can involve genetic, environmental, psychological, and biochemical factors [24]. Long-term deficits and lower quality of life for patients following ECMO compared with age-matched controls were reported for physical function [25,26,27,28]. The rate of depression following ECMO was reported to range from 42% [29] to as high as 50% [30]. Murphy et al. reported that patients who experienced a cardiac event continued to have depressive symptoms 12 months after the event [31]; however, the study did not compare rates of depression to the general population.

Depression not only affects social interactions and work but also reduces the motivation of patients to continue to do physical rehabilitation, preventing most patients from returning to their original jobs [22]. Therefore, the inability to return to previous social and work activities for patients who have survived ECMO and have been discharged from the hospital significantly impacts these patients’ mental health. Studies examined the impact of ECMO for survivors following hospital discharge on physical status and overall quality of life [26,27]. However, VA-ECMO and VV-ECMO differ in nature and, therefore, whether differences in ECMO treatments affect the post-discharge life of patients, particularly regarding physical and psychological health status, are worthy of investigation. Thus, the purpose of this study was to examine whether there were differences in depression status following hospital discharge between patients who had received VA-ECMO and those who received VA-ECMO.

## 2. Materials and Methods

### 2.1. Study Design

This study adopted a descriptive, cross-sectional, and correlational research design.

### 2.2. Participants and Setting

Participants were recruited via convenience and quota sampling. The sample comprised patients ≥20 years of age who had been discharged from the hospital at least one month after receiving ECMO at a medical center in northern Taiwan from June 2006-June 2020. From September 2019-August 2020, attending doctors at the outpatient clinic of the cardiovascular department provided the researchers with a list of 452 patients who had undergone ECMO between May 2016 and June 2020; of these, 307 had died and three had not yet been discharged from the hospital. The researchers reviewed the charts of the remaining 142 patients to determine whether they met the following inclusion criteria: (1) received ECMO and had been discharged from the hospital; (2) age ≥20 years; and (3) received a diagnosis requiring treatment with ECMO, such as cardiogenic shock related to heart disease, acute respiratory distress syndrome induced by lung disease, and extracorporeal cardio-pulmonary resuscitation (ECPR). Exclusion criteria for patients were difficulty with verbal expression, cognitive dysfunction, or lower limb disability prior to ECMO. A total of 104 patients met inclusion criteria; researchers were able to contact 97 of these patients. During the phone call, the researchers introduced themselves and explained the design and purpose of the study. A total of 90 patients agreed to participate (VA, *n* = 75; VV, *n* = 15); seven declined. An appointment was made at a time convenient to participants to meet at the outpatient clinic of the cardiovascular department. The flowchart for the selection of these 90 participants is shown in Figure 1.

To increase the amount of data for both modes of ECMO at a ratio close to 1:1, data from patients who had received VV-ECMO (*n* = 52) between April 2006 and April 2016 were included. These data were collected between October 2015-October 2016 using the same criteria as the 90 patients included in the analysis. A flowchart for the selection of these 52 participants is shown in Figure 2.

### 2.3. Data Collection

All data were collected from medical charts and questionnaires through face-to-face interviews after patients provided signed informed consent. The collection process was identical for the 90 participants from 2019-2020 and the 52 participants from 2015-2016.

### 2.4. Measures

#### 2.4.1. Demographic and Clinical Characteristics

Demographic data were collected using a survey questionnaire, which included gender, age, employment status, workload since ECMO, and self-perceived health status. Participants were asked whether their workload (work at home or their place of employment) since receiving ECMO was less, the same, or greater. Disease-related characteristics were obtained from a review of the patient’s chart, which included the mode of ECMO; disease severity, as indicated by the score on the Sequential Organ Failure Assessment (SOFA); and length of hospitalization.

#### 2.4.2. Hospital Anxiety and Depression Scale

The outcome variable of post-discharge depression status was measured with the subscale score on the Hospital Anxiety and Depression Scale (HADS-D) developed by Zigmond and Snaith [32]. The depression subscale comprises seven items that are statements about how frequently a participant has been feeling certain emotions over the past week, which is scored on a 4-point Likert scale from 0 (not at all) to 3 (most of the time); total scores range from 0–21 points, with higher scores indicating a greater frequency of symptoms of depression. HADS-D scores of 0–7 suggest no depression, scores of 8–10 indicate possible symptoms of depression, and scores of 11 or greater indicate the presence of depression. The scale has been applied in studies of other cardiopulmonary diseases, with good internal consistency, test–retest reliability, and concurrent validity; Cronbach’s alpha ranged from 0.7–0.85 [33,34]. This study used the Chinese version of the HADS-D, developed by Wang et al. [35], which was shown to be a reliable and valid screening instrument for the assessment of anxiety and depression in Chinese-speaking patients with coronary heart disease, with a Cronbach’s alpha of 0.79. In this study, Cronbach’s alpha was 0.66.

### 2.5. Statistical Analysis

Data were analyzed with the statistical software package SPSS 22.0 for Windows. The final analysis was conducted on data representing 142 patients who had received ECMO (VA, *n* = 75; VV, *n* = 67). Post hoc analysis showed that the power reached 0.98, which was greater than the commonly recommended 0.80, and allowed for the detection of differences exhibited by adult patients weaned from ECMO who survived to hospital discharge. Differences in demographics, disease-related data, and post-discharge depression status between VV-ECMO and VA-ECMO participants were analyzed with an independent t-test and chi-squared test. Variables that might affect the post-discharge depression status were identified with an independent t-test, one-way analysis of variance (ANOVA), and Pearson’s correlation. Significant variables were entered into multiple regression analysis, controlling for moderating variables for depression scores. Following input, “mode of ECMO” was added to explore differences in depression status between the two modes, under the control of moderating variables.

## 3. Results

### 3.1. Characteristics of Participants between Modes of ECMO

Characteristics of participants who received VV-ECMO and VA-ECMO are shown in Table 1. Several demographic variables differed between groups: participants treated with VA-ECMO were significantly older at the time of the survey (55.8 years ± 12.2) compared with those treated with VV-ECMO (50.3 ± 14.2 years, *p* = 0.015), and were more likely to be unemployed (64.0% vs. 41.8%, respectively; *p* = 0.027). The number of comorbidities at the time of the survey was also significantly greater for VA-ECMO (*p* = 0.016). Several clinical characteristics differed significantly between participants who received VV- compared with VA-ECMO: the SOFA score was higher (11.1 ± 2.8 versus 9.0 ± 2.6, respectively; *p* < 0.001); the numbers of days on ECMO (10.6 ± 9.5 versus 4.8 ± 3.4, respectively; *p* < 0.001) and mechanical ventilation (28.3 ± 24.3 and 18.0 ± 19.6, respectively; *p* = 0.006, respectively) were greater; and the number of days to hospital discharge was greater (*p* < 0.001). HADS-D scores for participants treated with VV-ECMO (6.0 ± 3.7) were significantly higher compared with VA-ECMO patients (4.4 ± 4.1, *p* = 0.018). Scores for the HADS-D suggested the possibility of depression in 16.4% of VV-ECMO patients and 6.7% of VA-ECMO patients. Scores for the HADS-D suggested the presence of depression in 12.9% of VV-ECMO patients and 13.3% of VA-ECMO patients. The time of discharge from the hospital for participants who received VV-ECMO did not differ significantly for mean HADS-D scores (*p* = 0.606).

### 3.2. Mode of ECMO as a Predictor of Post-Discharge Depression Status

Relationships between characteristics of all study participants and HADS-D scores were evaluated to determine whether there were any moderating variables (Table 2).

Three variables were significantly related to the HADS-D scores post-discharge: employment status, workload, and self-perceived health status. Therefore, these variables were entered with “mode of ECMO” into the multiple regression analysis (Table 3). After controlling for these moderating variables, the mode of ECMO was determined to be associated with post-discharge depression status for participants (*p* = 0.008). Post-discharge HADS-D scores were greater for participants treated with VV-ECMO (B = −1.707).

## 4. Discussion

### 4.1. Mode of ECMO and Differences in Patient Characteristics

Several demographic variables differed between participants treated with VA-ECMO compared with those who receive VV-ECMO. We found participants treated with VA-ECMO were significantly older than those who received VV-ECMO, which contrasted with two studies that reported patients treated with VA-ECMO were younger than those treated with VV-ECMO [36,37]. Patients included in these two studies were exclusively treated with ECMO for ARDS, whereas this study enrolled patients who received ECMO due to cardiogenic shock, ARDS, and ECPR, which is a more accurate reflection of the real-life situation for most hospitals. However, a study by Guttendorf et al. compared patients who received ECMO due to cardiac and respiratory problems, and the mean age of the VA-ECMO patients was greater than the VV-ECMO patients [8], which is consistent with our findings.

We could find no studies that compared post-discharge work status for survivors of VV-ECMO with survivors of VA-ECMO. However, the post-discharge work status for all ECMO patients varied with the length of time since the day of discharge. The proportion of patients employed less than 6 months post-discharge was reported to range from 8 to 31% [24], 51 to 69% at 7–12 months post-discharge [24,38], and seems to stabilize at 76% at 3 years post-discharge, as it ranges from 71.4–78% at 3–5 years post-discharge [39].

In this study, the proportion of employed patients was 49.3% after VV-ECMO and 32.0% after VA-ECMO. The association of lower employment post-discharge for patients who received VA-ECMO may be due to the placement of the catheter into the femoral artery, which increases the risk of lower limb complications [2]. Older age at the time of treatment was also associated with lower employment following hospital discharge, which may be because their family members wanted them to concentrate their efforts on health and recovery [23]. In addition, older adults with chronic heart disease are more likely to suffer from more comorbidities, which also might contribute to a failure in returning to work [40].

In this study, the severity of disease for patients who received VV-ECMO was significantly higher than patients in the VA-ECMO group. One possible reason for this finding was that this study enrolled VV-ECMO patients from 2006 to 2020, and the advances in medical devices and technology over this period might have affected the prognosis of patients. A recent study suggested that patients should receive ECMO as soon as possible to effectively improve the survival rate [41]. The indications for receiving ECMO have become more specific, which enables clinicians to provide treatment at the optimal time and prevents worsening of disease severity before administering treatment. Subsequently, participants in this study who received VV-ECMO spent significantly more days on ECMO than VA-ECMO patients (*p* < 0.001). This finding is similar to previous studies [8,36,37] and is associated with the type of diagnosis and severity of disease of patients.

Our findings regarding the number of days on mechanical ventilation differed from a previous study, which compared modes of ECMO only for patients with ARDS [37]. In that study, patients who received VA-ECMO had higher disease complexity due to complications from cardiac problems; therefore, the number of days on mechanical ventilation was higher compared with patients treated with VV-ECMO. In contrast, this study did not enroll patients with a single disease, the disease severity of patients that received VV-ECMO was higher, and the number of days on mechanical ventilation was greater.

### 4.2. Modes of ECMO and Differences in Depression Post-Discharge

Our findings showed higher scores for depression post-discharge for patients who received VV-ECMO compared with participants who received VA-ECMO. While it is generally believed that depression in critically ill patients after discharge is mainly related to cognitive problems, some studies suggest that somatic symptoms associated with lengthy hospitalization in the ICU, such as weakness, fatigue, and pain, are the critical factors leading to depression [23,42]. Patients often experience significant muscle loss [43] following the administration of high levels of sedation to ensure immobilization and avoid slippage of the ECMO catheter [44]. This also reduces the thickness and strength of the quadriceps muscle pre-ECMO compared with post-ECMO [45]. Patients typically continue to feel weak or easily fatigued after treatment is terminated, which contributes to limited daily life activities, and prevents them from returning to the work [24].

The higher scores for depression for survivors of VV-ECMO in our study might also be related to the greater severity of disease, number of days being treated with ECMO, and number of days on mechanical ventilation. Patients with higher severity of disease face a higher risk of impaired physical function, which is related to longer treatment time with ECMO and time on mechanical ventilation, which also increases the use of sedation and the number of days a patient is bedridden [46]. The use of sedatives is one of the risk factors for depression [47], and the increase in the time a patient is bedridden causes more muscle loss and greater severity of somatic symptoms, which increases the probability of subsequent depression. Moreover, the indications for modes of ECMO are different. Most patients who received VV-ECMO due to respiratory failure were induced by ARDS, thus, their lung function impairment was extremely serious. During treatment, 69.6% of such patients experienced ground-glass opacities in their lungs. After discharge, 70% of patients experienced fatigue, 50% had impaired physical functioning, and 36% had depression. Moreover, 46% reported both impaired physical function and fatigue, which affected daily life [48,49].

### 4.3. Incidence of Depression Post-Discharge

Our results indicated that 20% of participants who had undergone VA-ECMO had symptoms of depression 440 days after hospital discharge. This persistence in depressive symptoms was consistent with previous studies by Muller et al. in which the median time from hospital discharge was 32 months [50] and a study by Mirabel et al. who found patients continued to experience depressive symptoms 17 months after discharge [51]. Our finding demonstrating 28.4% of participants who had undergone VV-ECMO had symptoms of depression 1015 days after hospital discharge (33.8 months) was consistent with the findings of Sanfilippo and colleagues [29] in which patients 42% of patients reported depressive symptoms 32.4 months following hospital discharge.

Patients with ARDS and other critical illnesses face numerous stressors. Survivors of ARDS and others with long periods of hospitalization often have long recovery periods and high levels of psychological distress [17]. In a 5-year prospective longitudinal cohort study of 186 ARDS survivors, 32% of survivors were affected by depression [17]. Most participants in our study who underwent VV-ECMO were survivors with ARDS. Healthcare professionals need to identify risk factors for prolonged psychiatric morbidity in these survivors of ARDS, which were shown to include lower socioeconomic status, prior psychiatric morbidity, and worse baseline physical functioning [17].

Pharmacotherapy, such as serotonin reuptake inhibitors (SSRIs), in combination with psychotherapy, is generally regarded as the first line of treatment for patients with anxiety and depressive disorders [52]. Although it remains unclear whether endogenous kynurenines contribute directly to the initiation of neuropathological changes, it is well-established that the dysregulation of the metabolic kynurenine pathway is involved in the pathophysiology of several inflammation-linked neuropsychiatric diseases. However, manipulation of the kynurenine pathway through the administration of enzyme inhibitors may serve as a therapeutic strategy for neuropsychiatric disease [18,53]. New evidence for the implementation of new treatments includes the effectiveness of non-invasive brain stimulation (NIBS), which interferes with and modulates the abnormal activity of neural circuits via the amygdala–mPFC–hippocampus pathway, which is involved in the acquisition and consolidation of fear memories that are altered in many psychiatric disorders including depression [54]. Depression was associated with a decrease in dendritic spine density and dendritic complexities. Furthermore, NIBS techniques are now used to treat depression, modulate spine density, and dendritic complexities [55,56]. Existential phenomenological psychotherapy (EPP) was demonstrated to augment pharmacological support for patients with mood disorders for nearly a century, which helps patients find meaning and purpose in life to override physical limitations [52]. Therefore, we suggest early evaluations of patients who have undergone ECMO to enable timely support strategies that may combine pharmacological, NIBS, and psychological support.

Limitations on the activities of daily living are factors that can contribute to depression in patients after discharge from the hospital because they are forced to make lifestyle changes [40]. Limitations in activities of daily living combined with severe physical disabilities post-discharge can contribute to post-intensive care syndrome (PICS), which can include both depression and posttraumatic stress disorder (PTSD). Jackson et al. found depression to be far more common than PTSD for patients who had a critical illness in the ICU [42]. A longitudinal study of the health status of patients who received ECMO in Taiwan showed that the severity of depression in patients treated with VV-ECMO was higher than patients treated with VA-ECMO before discharge and this difference in depression was still present at 1 year post-discharge [22]; however, the sample size was small (*N* = 32). Although the two modes of ECMO have different indications, cannulation procedures, and accompanying complications, differences in disease-related characteristics between patients receiving VV-ECMO and VA-ECMO were not associated with depressive status after discharge. Thus, our large sample size in this study strengthened the association between greater levels of depression post-discharge for patients who had undergone VV-ECMO than for VA-ECMO patients.

The mental health of all ECMO patients is worthy of the attention of healthcare professionals. Therefore, we strongly encourage researchers in the fields of neuroscience and neuropsychiatry to examine the neurophysiological impact of ECMO on areas of the prefrontal cortex, which has been shown to be involved in depression [57]. Second, the use of NIBS techniques should be investigated as a valid alternative in the treatment of those patients that do not respond to psychotherapy and/or drug treatments [58].

### 4.4. Limitations

There were some limitations in this study. First, the initial sample size of VV-ECMO participants was relatively small, thus we included an additional 52 participants who had undergone VV-ECMO between June 2006 to April 2016. ECMO treatment during this earlier period may have differed, which may have had different impacts on depression for these participants compared with those who received ECMO between 2016 and 2020. We did not analyze differences between participants for these two time periods, which might have limited the strength of our findings.

Second, the Cronbach’s alpha of 0.66 showed that the item intercorrelations in this study were weak [59]. Although HADS-D was validated and shown to have good internal consistency reliability [33,34,35], studies with ECMO participants have not been reported [29,30,31,41]. Cronbach’s alpha is dependent upon the resulting distribution of scores’ variance; that is, the higher the value of the total variance, the greater the alpha value obtained [59]. In this study, the mean score of 6.0 (SD ± 3.7) for HADS-D was higher for participants receiving VV-ECMO compared with 4.4 (SD ± 4.1) for participants receiving VA-ECMO. The Cronbach’s alpha of HADS-D with participants receiving ECMO ought to be further examined in future studies.

Two additional limitations involve the design of the study. The cross-sectional design prevented determining a cause and effect between depression and the mode of ECMO treatment. Recruitment of participants via convenience sampling from one medical center, which is a nonprobabilistic sampling method, limited the generalizability of our findings to a broader sample of patients, which could result in bias and render the samples less representative. Therefore, our study findings cannot be extended to other patients receiving ECMO. Sample representativeness could be evaluated by comparing the mean values of HADS-D with ECMO participants through the meta-analysis of ECMO patients with depression [60].

Finally, there was a large gap between the number of patients enrolled that received VV-ECMO and those that received VA-ECMO. Thus, when data collection and comparisons were performed, there was a significant difference in the day of discharge between the two groups of patients. As depression can change over time and is susceptible to major life events, these factors might influence the research results.

## 5. Conclusions

With the advances in medical treatments, the indicators for evaluating whether a treatment is effective should be more than the survival rate; attention should also be paid to a patient’s prognosis. This study found that 20–28.4% of survivors of EMCO had HADS-D scores suggestive of the presence of depression. The mode of ECMO was associated with depression, and the depression status of VV-ECMO patients was more severe compared with VA-ECMO. Therefore, more clinical attention should be paid to evaluating the physical and mental function and social support of patients receiving VV-ECMO due to acute respiratory failure.

Thus, from the moment patients are successfully weaned from ECMO, it is recommended that continuous assessments and treatments of a patient’s mental health be conducted. A multi-professional clinical care path for patients that is established as soon as practically possible should include physicians, psychologists, neurologists, case managers, nurses, and pulmonary and cardiac physiotherapists. Prior to hospital discharge, a case manager should be assigned to the patient to provide information relative to their clinical needs for managing their health post-discharge. We suggest that healthcare professionals develop a partnership with patients and provide them with information on relaxation techniques, such as breathing control, mediation, or listening to their favorite music. A strong relationship between the patient and healthcare professionals could facilitate the initiation of self-management interventions for chronic diseases and encourage the patient to establish their own physical and pulmonary rehabilitation targets. Bandura’s social cognitive theory [61] suggests self-efficacy plays an important role in determining which self-management activities a person will perform, where the expectations of personal efficacy are based on four major sources of information: performance accomplishments, vicarious experience, verbal persuasion, and physiological states, which can be augmented by feedback from healthcare professionals. Finally, case managers should continue to follow up on the patient’s recovery in a timely manner and provide them with strategies for reducing psychological stress, such as pharmacological, NIBS, and/or EPP therapy, to help them return to a healthy lifestyle as soon as possible.

## Figures and Tables

**Figure 1 ijerph-19-03333-f001:**
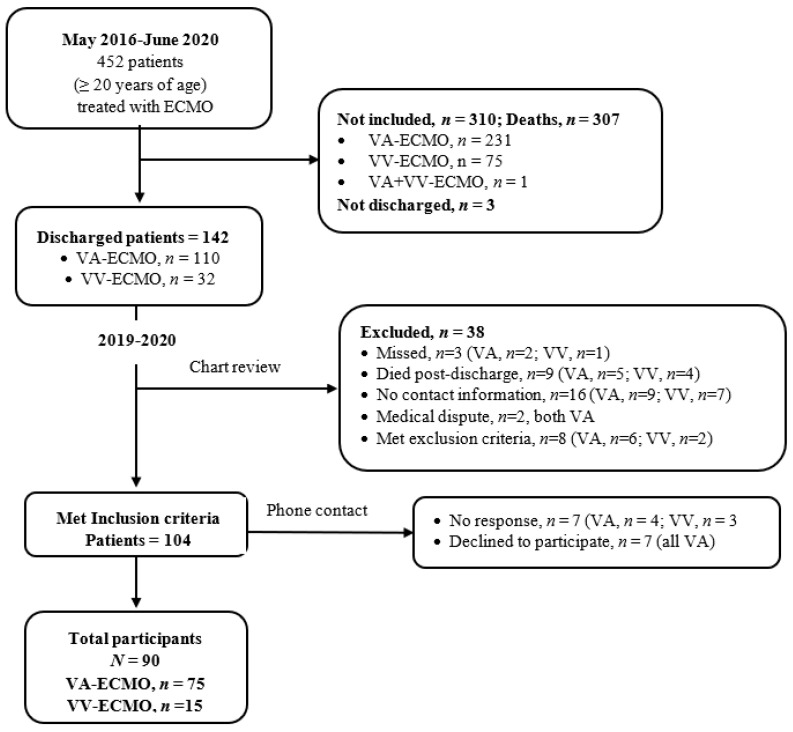
Flow chart of participant selection for data collected 2019–2020.

**Figure 2 ijerph-19-03333-f002:**
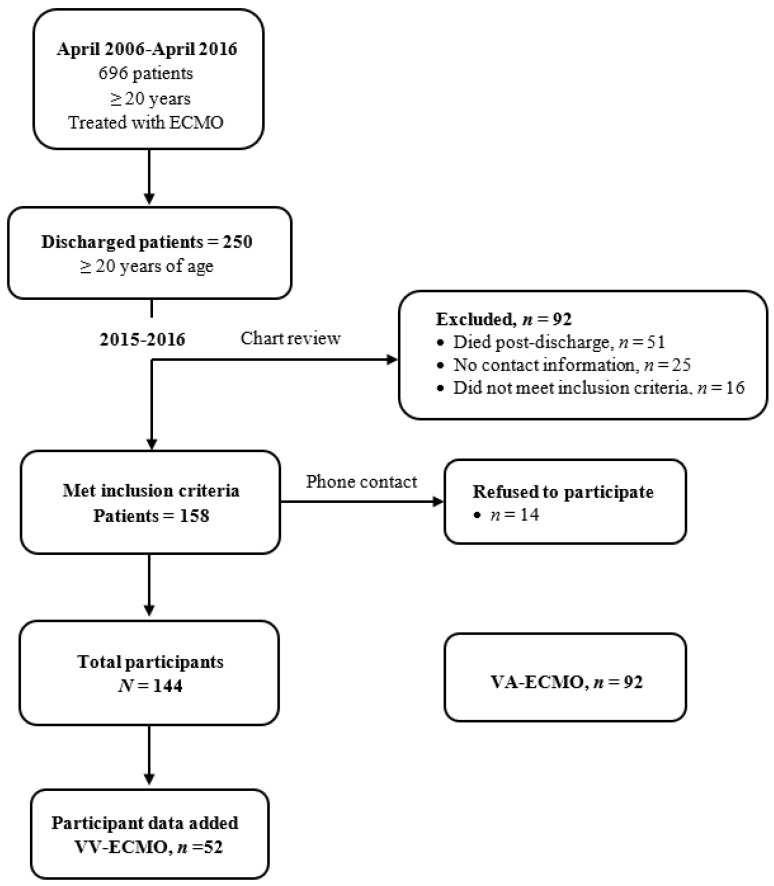
Flow chart of participant selection for data collected for VV-ECMO patients in 2015–2016.

**Table 1 ijerph-19-03333-t001:** Differences in characteristics of participants receiving veno-venous (VV) or veno-arterial (VA) extracorporeal life support (ECMO).

	VV-ECMO ^1^ (*n* = 67)		VA-ECMO ^2^ (*n* = 75)		t/χ^2^	*p*
Variables	*n* (%)	Mean ± SD ^3^	*n* (%)	Mean ± SD		
**Demographic characteristics at the time of treatment**						
Age, years		46.7 ± 14.3		54.2 ± 11.9	−3.368	0.001 **
**Gender**					0.070	0.791
Male	47 (70.1)		50 (66.7)			
Female	20 (29.9)		25 (33.3)			
**Disease-related characteristics at the time of treatment**						
SOFA ^5, b^		11.1 ± 2.8		9.0 ± 2.6	4.585	<0.001 ***
Length of hospitalization, days		58.4 ± 45.6		54.7 ± 46.8	0.477	0.634
Duration of ECMO, days		10.6 ± 9.5		4.8 ± 3.4	4.725	<0.001 ***
Duration of mechanical ventilation, days		28.3 ± 24.3		18.0 ± 19.6	2.819	0.006 **
**Number of complications**					3.930	0.269
0	23 (34.3)		17 (22.7)			
1	17 (25.4)		16 (21.3)			
2	13 (19.4)		18 (24.0)			
≥3	14 (20.9)		24 (32.0)			
**Demographic characteristics at the time of the survey**						
Age, years		50.3 ± 14.2		55.8 ± 12.2	−2.472	0.015 *
**Employment status**					7.257	0.027 *
Unemployed	28 (41.8)		48 (64.0)			
Employed	33 (49.3)		24 (32.0)			
Other (housekeeper/student)	6 (9.0)		3 (4.0)			
**Workload compared to pre-ECMO ^a^**					2.771	0.250
Unchanged	20 (29.9)		32 (42.7)			
Lighter	44 (65.7)		39 (52.0)			
Heavier	3 (4.5)		4 (5.3)			
**Disease-related characteristics at the time of the survey**						
Number of days since discharge		1015.1 ± 831.1		440.8 ± 322.5	5.310	<0.001 ***
Self-perceived health status		7.1 ± 1.9		7.2 ± 1.6	−0.463	0.644
BMI ^4^, kg/m^2^		25.6 ± 5.6		24.4 ± 3.5	1.505	0.135
**Number of comorbidities**					10.349	0.016 *
0	23 (34.3)		12 (16.0)			
1	16 (23.9)		12 (16.0)			
2	10 (14.9)		17 (22.7)			
≥3	18 (26.9)		34 (45.3)			
**Depression (HADS-D ^6^ score)**		6.0 ± 3.7		4.4 ± 4.1	2.387	0.018 *
0–7	48 (71.6)		60 (80.0)			
8–10	11 (16.4)		5 (6.7)			
11–21	8 (12.0)		10 (13.3)			
**Mean HADS-D score at the time of hospital discharge**						
April 2006 to April 2016	52 (77.6)	6.1 ± 3.6				
May 2016 to June 2020	15 (22.4)	5.5 ± 4.0				

^1^ Veno-venous extracorporeal membrane oxygenation; ^2^ veno-arterial extracorporeal membrane oxygenation; ^3^ standard deviation; ^4^ body mass index; ^5^ sequential organ failure assessment; ^6^ hospital anxiety and depression scale; ^a^ workload was defined as the participant’s current work at home or at their place of employment compared with pre-ECMO; ^b^ SOFA, VV, *n* = 62; VA, *n* = 75; * *p* < 0.05; ** *p* < 0.01; *** *p* < 0.001.

**Table 2 ijerph-19-03333-t002:** The relationship between the characteristics of all participants (*N* = 142) and HADS-D ^1^ depression scores after hospital discharge.

Variable	*n*	Mean ± SD ^2^	t/F	r	*p*
**Age at the time of treatment**	142	50.7 ± 13.6		0.032	0.078
VV-ECMO	67			0.017	0.89
VA-ECMO	75			0.165	0.157
**Age at the time of the survey**	142	53.2 ± 13.4		0.042	0.619
VV-ECMO	67			0.12	0.926
VA-ECMO	75			0.157	0.178
Gender			0.655		0.514
Male	97	5.0 ± 3.9			
Female	45	5.5 ± 4.1			
**Employment status**			6.293 ^a^		0.006 **
Unemployed	76	6.1 ± 4.5			
Employed	57	3.8 ± 2.9			
Housekeeper/student	9	5.2 ± 2.6			
**Workload compared to pre-ECMO ^3, b^**			6.006		0.003 **
Unchanged	52	3.7 ± 3.1			
Lighter	83	6.0 ± 4.2			
Heavier	7	6.3 ± 3.5			
BMI ^4^, kg/m^2^		5.0 ± 3.9		−0.013	0.882
Self-perceived health status at the time of the survey				−0.359	<0.001 ***
**Number of comorbidities at the time of the survey**			0.646		0.587
0	35	4.5 ± 3.4			
1	28	5.0 ± 4.4			
2	27	5.2 ± 4.0			
≥3	52	5.7 ± 3.9			
SOFA ^5, c^				0.049	0.572
Length of hospitalization, days				0.150	0.075
Duration of ECMO, days				0.065	0.442
Duration of mechanical ventilation, days				0.083	0.324
Number of days since discharge				0.031	0.710
**Number of complications**			1.283		0.283
0	40	4.4 ± 4.0			
1	33	4.9 ± 3.0			
2	31	6.1 ± 4.9			
≥3	38	5.4 ± 3.9			

^1^ Hospital anxiety and depression scale, ^2^ standard deviation; ^3^ extracorporeal membrane oxygenation; ^4^ body mass index; ^5^ sequential organ failure assessment; ^a^ Welch F; ^b^ workload was defined as the participant’s current work at home or their place of employment compared with pre-ECMO; ^c^ SOFA, VV, *n* = 62; VA, *n* = 75; ***p* < 0.01; *** *p <* 0.001.

**Table 3 ijerph-19-03333-t003:** Predictors of depression (HADS-D ^1^ score) for survivors of ECMO after hospital discharge (*N* = 142).

		Unstandardized		Standardized			
Variables	F	B	SE	*β*	t	*p*	Adjust R^2^
	6.863					<0.001 ***	0.200
(Intercept)		10.661	1.545		6.900	<0.001	
**Mode of ECMO ^2^**							
VV ^3^		(Reference)					
VA ^4^		−1.707	0.630	−0.217	−2.710	0.008 **	
Self-perceived health status at the time of the survey		−0.619	0.181	−0.272	−3.429	0.001**	
**Workload compared to pre-ECMO ^a^**							
Unchanged		(Reference)					
Lighter		0.832	0.707	0.104	1.177	0.241	
Heavier		1.877	1.440	0.103	1.304	0.194	
**Employment status**							
Unemployed		(Reference)					
Employed		−1.788	0.701	−0.223	−2.551	0.012 *	
Housekeeper/student		−0.469	1.314	−0.029	−0.357	0.721	

^1^ Hospital anxiety and depression scale; ^2^ extracorporeal membrane oxygenation; ^3^ veno-venous; ^4^ veno-arterial; ^a^ workload was defined as the participant’s current work at home or their place of employment compared with pre-ECMO; * *p* < 0.05; ** *p* < 0.01; *** *p* < 0.001.

## Data Availability

The data presented in this study are available on request from the corresponding author. The data are not publicly available due to privacy restrictions.

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
