# Peer review of "Post-Discharge Depression Status for Survivors of Extracorporeal Membrane Oxygenation (ECMO): Comparison of Veno-Venous ECMO and Veno-Arterial ECMO"

_ijerph, 2022, doi:10.3390/ijerph19063333_

Round 1
Reviewer 1 Report
I commend the authors for their study titled, “Post-Discharge Depression Status for Survivors of Extracorporeal Membrane Oxygenation (ECMO): Comparison of Veno-ve- 3 nous ECMO and Veno-arterial ECMO”
Here are my comments:
- The sample size is relatively small: 142 patients who had received ECMO (VA, n = 75; VV, n = 67).
- The study does not establish the actual cause of the differences in post -VV-ECMO depression versus pos-VA-ECMO depression, which could be related more to the clinical problems/indications for the procedures than the procedures themselves
- Comparison of post-VV-ECMO and post-VA-ECMO statuses does not constitute an apple-to-apple comparison. These are different procedures with different indications and possible different procedure-specific and shared complications with varying degrees of severity and subsequent health-related quality of life.
- VA-ECMO Provides cardiac support to assist systemic circulation, Requires arterial and venous cannulation; Bypasses pulmonary circulation/decreases pulmonary artery pressures, Could be used in RV failure, and the ECMO circuit connected in parallel to the heart and lungs. VV-ECMO Does not provide cardiac support to assist systemic circulation, requires only venous cannulation, cannot be used in RV failure, and the ECMO circuit is connected in series to the heart and lungs.
- The conclusion of the study appears to consist of a different narrative not supported by the research findings. None of the following author's perspectives appear to be supported by the results of this study although these could be part of routine patient care: the narrative about monitoring, assessment, treatment, the prognosis of patients, teamwork, care-provider/patient collaboration, and information on patient relaxation techniques/meditation /music, and follow up on patient recovery.
- In Sum, there is no new knowledge produced by this study that could influence existing clinical decision-making and improve upon established procedural practices.
References
Kurniawati, E. R., Vranken, N. P. A., Delnoij, T. S. R., Lorusso, R., van der Horst, I. C. C., Maessen, J. G., & Weerwind, P. W. (2021). Quality of life following adult veno-venous extracorporeal membrane oxygenation for acute respiratory distress syndrome: a systematic review. Quality of Life Research, 1-13.
Makdisi, G., & Wang, I. W. (2015). Extra corporeal membrane oxygenation (ECMO) review of a lifesaving technology. Journal of thoracic disease, 7(7), E166.
Munshi, L., Walkey, A., Goligher, E., Pham, T., Uleryk, E. M., & Fan, E. (2019). Venovenous extracorporeal membrane oxygenation for acute respiratory distress syndrome: a systematic review and meta-analysis. The Lancet Respiratory Medicine, 7(2), 163-172.
Author Response
Comment 1: The sample size is relatively small: 142 patients who had received ECMO (VA, n = 75; VV, n = 67).
Response 1: We appreciate the reviewer’s concern This cross-sectional retrospective study examined patients discharged from one hospital from June 2006 to June 2020. We believe the sample size is relatively large when compared with other retrospective studies of the depressive status of patients who had received ECMO. Our findings are the results of post hoc analysis, with a power of 0.98, which is much higher than the common recommendation of 0.80, which allowed for the detection of differences exhibited by adult patients weaned from ECMO who survived to hospital discharge. The table below lists previous studies and the sample size in which none are as large as our study.
Sample size and prevalence of anxiety and depression (HADS) for modes of ECMO in previous studies
|
|
|
|
Prevalence |
|
|
Reference |
N |
Mode of ECMO |
Anxiety |
Depression |
|
Mirabel (2011) |
26 |
VA (100%) |
50% |
35% |
|
2002-2009; median follow-up 3.6 y (range = 3 mo to 7 years) |
|
|
|
|
|
Bréchot (2013) |
9 |
VA |
55% |
11% |
|
2008-2011; evaluated Sept 2011 |
|
|
|
|
|
Muller (2016); |
41 |
VA |
34% |
20% |
|
2008-2103; evaluated 2014; questionnaires administered 18-54 mos. post discharge. |
|
|
|
|
|
Sanfilippo et al. (2019) |
33 |
VV |
39-42% |
42% |
|
2009-2016; evaluated September 2017 |
|
|
|
|
Bréchot N, Luyt C-E, Schmidt M, et al. 2013. Venoarterial Extracorporeal Membrane Oxygenation Support for Refractory Cardiovascular Dysfunction During Severe Bacterial Septic Shock. Critical Care Medicine, Jul;41(7): 1616-1626. doi: 10.1097/CCM.0b013e31828a2370
Mirabel M, Luyt CE, Leprince P, Trouillet JL, Léger P, Pavie A, et al. Outcomes, long‐term quality of life, and psychologic assessment of fulminant myocarditis patients rescued by mechanical circulatory support. Critical Care Medicine 2011;39(5):1029‐35. [PUBMED: 21336134]
Muller G, Erwan F, Lebreton et al. 2016. The ENCOURAGE mortality risk score and analysis of long-term outcomes after VA-ECMO for acute myocardial infarction with cardiogenic shock. Intensive Care Med. 2016 Mar;42(3):370-378. doi: 10.1007/s00134-016-4223-9. Epub 2016 Jan 29. PMID: 26825953
Sanfilippo, F.; Ippolito, M.; Santonocito, et al. Long-term functional and psychological recovery in a population of ARDS patients treated with VV-ECMO and in their caregivers. Minerva. Anestesiol. 2019, 85, 971-980. doi: 10.23736/S0375-9393.19.13095-7.
Comment 2: The study does not establish the actual cause of the differences in post -VV-ECMO depression versus post-VA-ECMO depression, which could be related more to the clinical problems/indications for the procedures than the procedures themselves.
Response 2: We agree with the reviewer that this study does not establish the cause of differences, which is not possible with a cross-sectional study. However, our aim was to examine if depression status differed between patients post-VV- ECMO and post-VA- ECMO under the control of moderating variables. Although other studies have examined depressive status post-ECMO, our findings fill a research gap on the impact of the mode of ECMO on participants’ mental health, which has not previously been explored.
Comment 3: Comparison of post-VV-ECMO and post-VA-ECMO statuses does not constitute an apple-to-apple comparison. These are different procedures with different indications and possible different procedure-specific and shared complications with varying degrees of severity and subsequent health-related quality of life.
Response 3: We thank the reviewer for this comment. We agree that the two modes of ECMO have different indications, cannulation procedures and accompanying complications. We also have found significant difference in disease-related characteristics between patients receiving the two forms of ECMO, as shown in Table 1. However, our correlational analysis showed these disease-related characteristics were not associated with depressive status after discharge. Chen et al. (2018) reported similar findings, but the sample size of the study was relatively small (N=32). We have added the following to the discussion (Pages 21-22):
A longitudinal study of the health status of patients who received ECMO in Taiwan showed depression in patients treated with VV-ECMO was higher than patients treated with VA-ECMO before discharge and this difference in depression was still present at 1-year post-discharge [13], however the sample size was small (N = 32). Although the two modes of ECMO have different indications, cannulation procedures and accompanying complications, differences in disease-related characteristics between patients receiving VV-ECMO and VA-ECMO forms of ECMO were not associated with depressive status after discharge. Thus, our large sample size in this study strengthens the association between greater levels of depression post-discharge for patients who have undergone VV-ECMO than for VA-ECMO patients.
[13] Chen, K. H., Chen, Y. T., Yeh, S. L., Weng, L. C., & Tsai, F. C. (2018). Changes in quality of life and health status in patients with extracorporeal life support: A prospective longitudinal study. PLOS ONE, 13(5), e0196778. https://doi.org/10.1371/journal.pone.0196778
Comment 4: VA-ECMO Provides cardiac support to assist systemic circulation, Requires arterial and venous cannulation; Bypasses pulmonary circulation/decreases pulmonary artery pressures, Could be used in RV failure, and the ECMO circuit connected in parallel to the heart and lungs. VV-ECMO Does not provide cardiac support to assist systemic circulation, requires only venous cannulation, cannot be used in RV failure, and the ECMO circuit is connected in series to the heart and lungs.
Response 4: The authors appreciate the reviewer’s concerns and questions. We have revised Introduction (Page 3-4) as follows:
There are two modes of ECMO, which differ according to medical indications. The Extracorporeal Life Support Organization (ESLO) recommends veno-arterial (VA-ECMO) for patients with refractory cardiogenic shock due to acute myocardial infarction, fulminant myocarditis, reactions to cardiotoxic drugs, end-stage dilated or ischemic cardiomyopathy, hypothermia with refractory cardiocirculatory instability, massive pulmonary embolism, or patients with transplant acute cardiogenic shock [3]. During VA-ECMO, the ECMO circuit is connected in parallel to the heart and lungs, which allows blood to bypass both organs and returned to the arterial system by peripheral cannulation [3,4]. Indicators for patients needing veno-venous ECMO (VV-ECMO) include severe, acute, reversible respiratory failure that is refractory to optimal medical management [5]. Venous cannulation is performed singly through the right internal jugular vein or with double venous cannulas placed in the common femoral vein and right internal jugular or femoral vein [4,5].
Comment 5: The conclusion of the study appears to consist of a different narrative not supported by the research findings. None of the following author's perspectives appear to be supported by the results of this study although these could be part of routine patient care: the narrative about monitoring, assessment, treatment, the prognosis of patients, teamwork, care-provider/patient collaboration, and information on patient relaxation techniques/meditation /music, and follow up on patient recovery.
Response 5: The authors appreciate the reviewer’s question. We revised the Conclusion (Page 23-24) as follows:
With the advances in medical treatments, the indicators for evaluating whether a treatment is effective should be more than the survival rate; attention should also be paid to a patient’s prognosis. This study found that 20-28.4% of survivors of ECMO had HADS-D scores suggestive of the presence of depression. The mode of ECMO was associated with depression, and the depression status of VV-ECMO patients was more severe compared with VA-ECMO. Therefore, more clinical attention should be paid to evaluating physical and mental function and social support of patients receiving VV-ECMO due to acute respiratory failure.
Thus, from the moment patients are successfully weaned from ECMO, it is recommended that continuous assessments and treatments of a patient’s mental health be conducted. A multi-professional clinical care path for patients that is established as soon as practically possible should include physicians, psychologists, neurologists, case managers, nurses, and pulmonary and cardiac physiotherapists. Prior to hospital discharge, a case manager should be assigned to the patient to provide information relative to their clinical needs for managing their health post-discharge. We suggest healthcare professionals develop a partnership with patients and provide them with information on relaxation techniques, such as breathing control, mediation or listening to their favorite music. A strong relationship between patient and healthcare professionals could facilitate initiation of self-management interventions for chronic diseases and encourage patients to establish their own physical and pulmonary rehabilitation targets. Bandura’s social cognitive theory [47] suggests self-efficacy plays an important role in determining which self-management activities a person will perform, with expectations of personal efficacy based on four major sources of information: performance accomplishments, vicarious experience, verbal persuasion, and physiological states, which can be augmented by feedback from healthcare professionals. Finally, case managers should continue to follow-up on the patient’s recovery in a timely manner and provide them with strategies for reducing psychological stress such as pharmacological or EPP therapy to help them return to a healthy lifestyle as soon as possible.
Comment 6: In Sum, there is no new knowledge produced by this study that could influence existing clinical decision-making and improve upon established procedural practices.
Response 6: We believe our findings would be of interest to readers of International Journal of Environmental Research and Public Health. Understanding factors that can contribute to poor outcomes for survivors of ECMO can help guide healthcare professionals in developing strategies and treatment plans to assist patients with the challenges of recovery prior to weaning from ECMO. A better understanding of the risks of ECMO-associated complications and outcomes specific to the mode of ECMO, could improve the mental health and quality of life for survivors of VV- and VA-ECMO.
Reviewer 2 Report
Dear Authors,
Extracorporeal membrane oxygenation (ECMO) provides prolonged cardiac and respiratory support to sustain life. The authors conducted a cross-sectional study to assess post-discharge depression in patients who received veno-venous (VV) and veno-arterial (VA) ECMO by the Hospital Anxiety and Depression Scale-Depression (HADS-D). The results showed that the scores are significantly higher in patients who received VV-ECMO than those with VA-ECMO and that the mode of ECMO is associated with depression with a greater level of post-discharge 33
depression for participants who received VV- ECMO. The authors concluded that more psychological care is recommended for patients who receive VV-ECMO.
Please consider the following:
- A graphical abstract is highly recommended.
- Page 1, Abstract: Please rephrase the second result to make it clear.
- Page 1, Keywords: Please add up to ten keywords.
- Pages 1-2, Introduction:
- Please describe more about indication and difference for VA-ECMO and VV-ECMO.
- In my opinion, I think that more information about depressive disorder would provide suitable background here. Thus, I suggest the authors to make an effort to provide a brief overview of the pertinent published literature that offer a perspective on definition, causes and symptoms of depression, because as it stands, this information is not highlighted in the text.
- Accordingly, I suggest the authors to reshape the Introduction section, which seems inhomogeneous and dispersive. Hence, I believe that it may be useful to have a general overview of psychiatric mood disorders (i.e., depression, anxiety and PTSD) and, specifically, of the related symptoms, diagnosis and cutting-edge treatments. For this reason, I would suggest some crucial references that will methodologically fit with the present manuscript, as new evidence for the implementation of new methods to treat such disorders (i.e., by the means of non-invasive brain stimulation techniques (NIBS)) in the treatment in mental disorder in humans: for example, a recent review by Borgomaneri and colleagues (2021, Neuroscience and Biobehavioral Reviews) described the potential and effectiveness of non-invasive brain simulation (NIBS) to interfere and modulate the abnormal activity of neural circuits (i.e., amygdala-mPFC-hippocampus) involved in the acquisition and consolidation of fear memories, which are altered in many psychiatric disorders (i.e., anxiety disorder, phobias, post-traumatic stress disorder or depression). Interestingly, another Borgomaneri and colleagues’ piece of evidence (2021, Journal of Affective Disorders) illustrated the therapeutic potential of NIBS as a valid alternative in the treatment of untypically persistent memories that characterized those patients that do not respond to psychotherapy and/or drug treatments. In addition, I would recommend citing relevant studies by Tanaka and colleagues (2013, Regulatory Peptides) and by Balogh and colleagues (2021, Biomedicines) on anti-depressant effects of neural circuitry mediating affective states.
- Page 2, Participants and Settings: Since patients were contacted with phone calls, could authors explain why they have decided to administer the HADS Scale over the Patient Health Questionnaire-9 (PHQ-9), a brief self-report diagnostic instrument for the diagnosis of depression, whose administration by telephone has been validated (see Pinto-Meza et al., 2005 - Journal of general internal medicine), making it the most reliable procedure for assessing depression over telephone?
- Page 5, Results: I suggest reorganizing this section for more clarity, providing full statistical details, to ensure in-depth understanding and replicability of the findings.
- Page 9-11, Discussion: In my opinion, this study would be more compelling and useful to a broad readership if the authors moved beyond discussing associations between modes of ECMO and differences in depression post-discharge and discussed theoretical and methodological avenues in need of refinement, using this evidence to suggest a path forward. In this regard, I would suggest, to have a more comprehensive and through overview on this topic, to also have more information on neural substrates of depression, particularly on frontal lobe dysfunction in depressed patients, from additional evidence and techniques that have examined the effects that functional alteration in the prefrontal cortex have on memory and learning in patients: in a recent study that involved patients with brain lesions, Battaglia and colleagues (2020, The Journal of Neuroscience) revealed that the ventromedial prefrontal cortex (vmPFC) is involved in the acquisition of emotional conditioning (i.e., learning), assessing the role of this region and how its disrupted function may contribute to irregular response to fear and, therefore, to the development of many psychiatric mood disorders (as anxiety and depression. Finally, in a recent review on vmPFC subregional contributions, Battaglia and colleagues (2021, Molecular psychiatry) discussed the role of vmPFC in processing safety-threat information and their relative value, and how this region is fundamental for the evaluation and representation of stimulus-outcome’s value needed to produce sustained physiological responses. I also recommend the review by Zhang and colleagues (2018, CNS neuroscience & therapeutics) and by Pizzagalli and Roberts (2022, Neuropsychopharmacology) that have focused on this topic.
- Regarding Table 1: I am confused by the dispersive presentation of demographic information in Table 1. Thus, I would ask the authors to reorganize data, providing a more structured and organized table, with evidenced rows and columns, to help the reader in getting all the information at one glance.
The manuscript contains two figures, three tables and 35 references, which I believe are dramatically insufficient for a research paper. The whole paper needs a quick process of revising the manuscript to improve readability: authors should include more evidence and reviews to back their claims and focus on deepening the subject of their manuscript, as the bibliography is too concise; also, there are a few typos and wrong verb tenses. Finally, the manuscript carries important value presenting the difference in the occurrence of post-discharge between VA-ECMO and VV-ECMO and I recommend this manuscript for publication after major revision.
Author Response
General comment: Extracorporeal membrane oxygenation (ECMO) provides prolonged cardiac and respiratory support to sustain life. The authors conducted a cross-sectional study to assess post-discharge depression in patients who received veno-venous (VV) and veno-arterial (VA) ECMO by the Hospital Anxiety and Depression Scale-Depression (HADS-D). The results showed that the scores are significantly higher in patients who received VV-ECMO than those with VA-ECMO and that the mode of ECMO is associated with depression with a greater level of post-discharge depression for participants who received VV- ECMO. The authors concluded that more psychological care is recommended for patients who receive VV-ECMO.
Comment 1: A graphical abstract is highly recommended.
Response 1: We appreciate the reviewer’s suggestion. We have added a graphical abstract (Page 3), which is shown below.
Comment 2: Page 1, Abstract: Please rephrase the second result to make it clear.
Response 2: We appreciate the reviewer’s suggestion. We revised the Abstract as follows:
In addition, the mode of ECMO was a predictor of post-discharge depression (p = .008) for participants who received VV- ECMO.
Comment 3: Page 1, Keywords: Please add up to ten keywords.
Response 3: Thank you for this suggestion. We have added additional keywords on Page 2 (in red font):
Keywords: cross-sectional design, extracorporeal membrane oxygenation, veno-arterial, veno-venous, cardiac failure, respiratory failure, depression status, mental health support
Comment 4: Please describe more about indication and difference for VA-ECMO and VV-ECMO.
Response 4: The authors appreciate the reviewer’s suggestion. We have revised the Introduction (Page 3-4) as follows:
There are two modes of ECMO, which differ according to medical indications. The Extracorporeal Life Support Organization (ESLO) recommends veno-arterial (VA-ECMO) for patients with refractory cardiogenic shock due to acute myocardial infarction, fulminant myocarditis, reactions to cardiotoxic drugs, end-stage dilated or ischemic cardiomyopathy, hypothermia with refractory cardiocirculatory instability, massive pulmonary embolism, or patients with transplant acute cardiogenic shock [3]. During VA-ECMO, the ECMO circuit is connected in parallel to the heart and lungs, which allows blood to bypass both organs and returned to the arterial system by peripheral cannulation [3,4]. Indicators for patients needing veno-venous ECMO (VV-ECMO) include severe, acute, reversible respiratory failure that is refractory to optimal medical management [5]. Venous cannulation is performed singly through the right internal jugular vein or with double venous cannulas placed in the common femoral vein and right internal jugular or femoral vein [4,5].
Comment 5: In my opinion, I think that more information about depressive disorder would provide suitable background here. Thus, I suggest the authors to make an effort to provide a brief overview of the pertinent published literature that offer a perspective on definition, causes and symptoms of depression, because as it stands, this information is not highlighted in the text.
Response 5: We have added more information about depression to the Introduction (Page 4-5) as follows:
Psychological problems are common in patients with heart failure [9], acute respiratory distress syndrome [10] and critical illness [11]. Many patients experience constant depressive symptoms 1-year post-discharge following ECMO due to physical complications [12]. However, after discharge following ECMO, patients often experience poorer quality of life than the general population, delays in physical recovery, and a risk of psychological problems [12-14]. Many patients experience constant depressive symptoms 1-year post-discharge following ECMO due to physical complications [12]. Depression is a neurological illness with persistent symptoms, which can interfere with an individual’s quality of life following discharge from ECMO [13]. Symptoms can include sadness, a decrease in activities one used to enjoy, changes in weight, and difficulty sleeping, which can involve genetic, environmental, psychological, and biochemical factors [15]. Long-term deficits and lower quality of life, for patients following ECMO, compared with age-matched controls, have been reported for physical function [16-19]. The rate of depression following ECMO has been reported to range from 42% [20] to as high as 50% [21]. Murphy et al. (2020) reported patients who experienced a cardiac event continued to have depressive symptoms 12-months after the event [22], however, the study did not compare rates of depression to the general population.
Comment 6: Accordingly, I suggest the authors to reshape the Introduction section, which seems inhomogeneous and dispersive. Hence, I believe that it may be useful to have a general overview of psychiatric mood disorders (i.e., depression, anxiety and PTSD) and, specifically, of the related symptoms, diagnosis and cutting-edge treatments. For this reason, I would suggest some crucial references that will methodologically fit with the present manuscript, as new evidence for the implementation of new methods to treat such disorders (i.e., by the means of non-invasive brain stimulation techniques (NIBS)) in the treatment in mental disorder in humans: for example, a recent review by Borgomaneri and colleagues (2021, Neuroscience and Biobehavioral Reviews) described the potential and effectiveness of non-invasive brain simulation (NIBS) to interfere and modulate the abnormal activity of neural circuits (i.e., amygdala-mPFC-hippocampus) involved in the acquisition and consolidation of fear memories, which are altered in many psychiatric disorders (i.e., anxiety disorder, phobias, post-traumatic stress disorder or depression). Interestingly, another Borgomaneri and colleagues’ piece of evidence (2021, Journal of Affective Disorders) illustrated the therapeutic potential of NIBS as a valid alternative in the treatment of untypically persistent memories that characterized those patients that do not respond to psychotherapy and/or drug treatments. In addition, I would recommend citing relevant studies by Tanaka and colleagues (2013, Regulatory Peptides) and by Balogh and colleagues (2021,Biomedicines) on anti-depressant effects of neural circuitry mediating affective states.
Response 6: The authors appreciate the reviewer’s suggestion. However, none of the authors involved in this manuscript have a background in neurology, therefore this is beyond the scope of our study. However, we have revised and reorganized the Introduction to better explain the importance of depression following ECMO (please see Responses 4 and 5, above). We have cited Balogh et al. [43] in the following revision of the Discussion (Page 20-21) as follows:
4.3 Incidence of Depression Post-discharge
Our results indicated 20% of participants who had undergone VA-ECMO had symptoms of depression 440 days after hospital discharge. This persistence in depressive symptoms is consistent with previous studies by Muller et al. (2013) in which the median time from hospital discharge was 32 months [41] and a study by Mirabel et al. (2011) who found patients continued to experience depressive symptoms 17 months after discharge [42]. Our finding demonstrating 28.4% of participants who had undergone VV-ECMO had symptoms of depression 1015 days after hospital discharge (33.8 months) is consistent with the findings of Sanfilippo and colleagues [20] in which patients 42% of patients reported depressive symptoms 32.4 months following hospital discharge. Pharmacotherapy, such as serotonin re-uptake inhibitors (SSRIs), in combination with psychotherapy is generally regarded as the first line of treatment for patients with anxiety and depressive disorders [43]. Existential phenomenological psychotherapy (EPP) has been demonstrated to augment pharmacological support for patients with mood disorders for nearly a century, which helps patients find meaning and purpose in life to override physical limitations [43]. Therefore, we suggest early evaluations of patients who have undergone ECMO to enable timely support strategies that include both pharmacological and psychological support.
We have also revised the Conclusions (Pages 23-24):
With the advances in medical treatments, the indicators for evaluating whether a treatment is effective should be more than the survival rate; attention should also be paid to a patient’s prognosis. This study found that 20-28.4% of survivors of ECMO had HADS-D scores suggestive of the presence of depression. The mode of ECMO was associated with depression, and the depression status of VV-ECMO patients was more severe compared with VA-ECMO. Therefore, more clinical attention should be paid to evaluating physical and mental function and social support of patients receiving VV-ECMO due to acute respiratory failure.
Thus, from the moment patients are successfully weaned from ECMO, it is recommended that continuous assessments and treatments of a patient’s mental health be conducted. A multi-professional clinical care path for patients that is established as soon as practically possible should include physicians, psychologists, neurologists, case managers, nurses, and pulmonary and cardiac physiotherapists. Prior to hospital discharge, a case manager should be assigned to the patient to provide information relative to their clinical needs for managing their health post-discharge. We suggest healthcare professionals develop a partnership with patients and provide them with information on relaxation techniques, such as breathing control, mediation or listening to their favorite music. A strong relationship between patient and healthcare professionals could facilitate initiation of self-management interventions for chronic diseases and encourage patients to establish their own physical and pulmonary rehabilitation targets. Bandura’s social cognitive theory [47] suggests self-efficacy plays an important role in determining which self-management activities a person will perform, with expectations of personal efficacy based on four major sources of information: performance accomplishments, vicarious experience, verbal persuasion, and physiological states, which can be augmented by feedback from healthcare professionals. Finally, case managers should continue to follow-up on the patient’s recovery in a timely manner and provide them with strategies for reducing psychological stress such as pharmacological or EPP therapy to help them return to a healthy lifestyle as soon as possible.
Comment 7: Page 2, Participants and Settings: Since patients were contacted with phone calls, could authors explain why they have decided to administer the HADS Scale over the Patient Health Questionnaire-9 (PHQ-9), a brief self-report diagnostic instrument for the diagnosis of depression, whose administration by telephone has been validated (see Pinto-Meza et al., 2005 -Journal of general internal medicine), making it the most reliable procedure for assessing depression over telephone?
Response 7: The authors appreciate the reviewer’s concerns and questions. We revised the Participants and Setting (Pages 6) as follows:
During the phone call the researchers introduced themselves and explained the design and purpose of the study. A total of 90 patients agreed to participate (VA, n = 75; VV, n = 15); seven declined. An appointment was made at a time convenient to participants to meet at the outpatient clinic of the cardiovascular department. The flowchart for selection of these 90 participants is shown in Figure 1.
HADS-D has good reliability and validity. It has been used in many studies of mental health outcomes with discharged patients weaned from ECMO. Therefore, the use of HADS-D could allow for our data to be included in future a meta-analysis.
Sample size and prevalence of anxiety and depression (HADS) for modes of ECMO in previous studies
|
|
|
|
Prevalence |
|
|
Reference |
N |
Mode of ECMO |
Anxiety |
Depression |
|
Mirabel (2011) |
26 |
VA (100%) |
50% |
35% |
|
2002-2009; median follow-up 3.6 y (range = 3 mo to 7 years) |
|
|
|
|
|
Bréchot (2013) |
9 |
VA |
55% |
11% |
|
2008-2011; evaluated Sept 2011 |
|
|
|
|
|
Muller (2016); |
41 |
VA |
34% |
20% |
|
2008-2103; evaluated 2014; questionnaires administered 18-54 mos. post discharge. |
|
|
|
|
|
Sanfilippo et al. (2019) |
33 |
VV |
39-42% |
42% |
|
2009-2016; evaluated September 2017 |
|
|
|
|
Bréchot N, Luyt C-E, Schmidt M, et al. 2013. Venoarterial Extracorporeal Membrane Oxygenation Support for Refractory Cardiovascular Dysfunction During Severe Bacterial Septic Shock. Critical Care Medicine, Jul;41(7): 1616-1626. doi: 10.1097/CCM.0b013e31828a2370
Mirabel M, Luyt CE, Leprince P, Trouillet JL, Léger P, Pavie A, et al. Outcomes, long‐term quality of life, and psychologic assessment of fulminant myocarditis patients rescued by mechanical circulatory support. Critical Care Medicine 2011;39(5):1029‐35. [PUBMED: 21336134]
Muller G, Erwan F, Lebreton et al. 2016. The ENCOURAGE mortality risk score and analysis of long-term outcomes after VA-ECMO for acute myocardial infarction with cardiogenic shock. Intensive Care Med. 2016 Mar;42(3):370-378. doi: 10.1007/s00134-016-4223-9. Epub 2016 Jan 29. PMID: 26825953
Sanfilippo, F.; Ippolito, M.; Santonocito, et al. Long-term functional and psychological recovery in a population of ARDS patients treated with VV-ECMO and in their caregivers. Minerva. Anestesiol. 2019, 85, 971-980. doi: 10.23736/S0375-9393.19.13095-7.
Comment 8: Page 5, Results: I suggest reorganizing this section for more clarity, providing full statistical details, to ensure in-depth understanding and replicability of the findings.
Response 8: We thank the reviewer for this suggestion. We have reorganized the results and added additional statistical details to the tables: Table 1, pages 11-13; Table 2, pages 14-15; and Table 3, page 17.
Comment 9: Page 9-11, Discussion: In my opinion, this study would be more compelling and useful to a broad readership if the authors moved beyond discussing associations between modes of ECMO and differences in depression post-discharge and discussed theoretical and methodological avenues in need of refinement, using this evidence to suggest a path forward. In this regard, I would suggest, to have a more comprehensive and through overview on this topic, to also have more information on neural substrates of depression, particularly on frontal lobe dysfunction in depressed patients, from additional evidence and techniques that have examined the effects that functional alteration in the prefrontal cortex have on memory and learning in patients: in a recent study that involved patients with brain lesions, Battaglia and colleagues (2020, The Journal of Neuroscience) revealed that the ventromedial prefrontal cortex (vmPFC) is involved in the acquisition of emotional conditioning (i.e., learning), assessing the role of this region and how its disrupted function may contribute to irregular response to fear and, therefore, to the development of many psychiatric mood disorders (as anxiety and depression. Finally, in a recent review on vmPFC subregional contributions, Battaglia and colleagues (2021, Molecular psychiatry) discussed the role of vmPFC in processing safety-threat information and their relative value, and how this region is fundamental for the evaluation and representation of stimulus-outcome’s value needed to produce sustained physiological responses. I also recommend the review by Zhang and colleagues (2018, CNS neuroscience & therapeutics) and by Pizzagalli and Roberts (2022, Neuropsychopharmacology) that have focused on this topic.
Response 9: We thank the reviewer for these comments and suggestions. As mentioned in our response to Comment 6, our group does not include a neurologist or neuropsychologist. The purpose of our study was to show that there are differences in depressive symptoms between survivors of VV- and VA-ECMO. The possible causes for these differences would be an important topic of investigation for researchers in the fields of neuroscience or neuropsychiatry. We have cited Pizzagalli and Roberts (2022) in the last paragraph of the discussion (Page 22) as follows:
The mental health of all ECMO patients is worthy of the attention of healthcare professionals. Therefore, we strongly encourage researchers in the fields of neuroscience and neuropsychiatry to examine the neurophysiological impact of ECMO on areas of the prefrontal cortex, which has been shown to be involved in depression [44].
Comment 10: Regarding Table 1: I am confused by the dispersive presentation of demographic information in Table 1. Thus, I would ask the authors to reorganize data, providing a more structured and organized table, with evidenced rows and columns, to help the reader in getting all the information at one glance.
Response 10: Thank you for this suggestion. We have revised the Table 1, which now identifies data collected after treatment and from the survey. We have added bold font to draw attention to each variable and aligned the rows and columns to improve readability of the data. Please see Pages 12-13.
Comment 11: The manuscript contains two figures, three tables and 35 references, which I believe are dramatically insufficient for a research paper. The whole paper needs a quick process of revising the manuscript to improve readability: authors should include more evidence and reviews to back their claims and focus on deepening the subject of their manuscript, as the bibliography is too concise; also, there are a few typos and wrong verb tenses. Finally, the manuscript carries important value presenting the difference in the occurrence of post-discharge between VA-ECMO and VV-ECMO and I recommend this manuscript for publication after major revision.
Response 11: We thank the reviewer for these comments. We have revised the paper to improve readability and the total number of references is now 47. However, it is not unusual for IJERPH to publish studies on ECMO with only 35 references (Oh et al., 2021; Krupa et al., 2021).

Reviewer 3 Report
Dear authors.
An interesting study although it would be interesting to know how this issue is in times of pandemic in which public health has significantly worsened the quality of care. If you can put some reference on this subject it would be very interesting and current.
Another point I wanted to emphasize is that perhaps it is not appropriate to use the term well-being or quality of life, when what is being measured is depressive symptoms.
Now, I am going to comment on some topics to improve your article.
INTRODUCTION
In the present study you study the differences in the effect of VA-ECMO and VV-69 ECMO treatments on patients' life after discharge. It would be interesting to know if similar studies have been done before. Please look for any meta-analysis or studies on the subject.
It is also important that you contextualize the country in which you have performed the study, and to know if the treatments are similar in the rest of the countries.
The methodology is well written
In the discussion, the authors highlight the limitations that are quite considerable in the present study. For example, the sample size.
In the conclusions, the analysis of the psychological treatments that can be given to these patients is somewhat poor. It should go more deeply into this subject and say which professionals should participate in the multidisciplinary treatment mentioned. It would also be interesting to mention whether psychological help is currently provided.
Thank you
Author Response
General comment: An interesting study although it would be interesting to know how this issue is in times of pandemic in which public health has significantly worsened the quality of care. If you can put some reference on this subject it would be very interesting and current.
Comment 1: Another point I wanted to emphasize is that perhaps it is not appropriate to use the term well-being or quality of life, when what is being measured is depressive symptoms.
Response 1: The authors appreciate the reviewer’s concerns and questions.
- Regarding the issue of the pandemic, this study was conducted prior to the appearance of COVID-19 in Taiwan. Taiwan’s experience with the coronavirus differs dramatically from the rest of the world due to the immediate response when the virus was first identified.
- Regarding the use of the term well-being or quality of life, our point was that this variable is commonly assessed for patients who have undergone ECMO, while our study’s aim was to examine if depression status differed between patients who post-VV- ECMO and post-VA- ECMO under the control of moderating variables. Status of well-being or quality of life was not our research purpose.
Comment 2: INTRODUCTION
In the present study you study the differences in the effect of VA-ECMO and VV- ECMO treatments on patients' life after discharge. It would be interesting to know if similar studies have been done before. Please look for any meta-analysis or studies on the subject.
Response 2: We thank the reviewer for this comment. Chen et al. (2018) compared the difference between VV-ECMO and VA-ECM but the sample size of the study was relatively small (N=32). There is no meta-analysis or systemic review about the mental health of adult ECMO patients following hospital discharge.
Chen, K. H., Chen, Y. T., Yeh, S. L., Weng, L. C., & Tsai, F. C. (2018). Changes in quality of life and health status in patients with extracorporeal life support: A prospective longitudinal study. PLOS ONE, 13(5), e0196778. https://doi.org/10.1371/journal.pone.0196778
Comment 3: It is also important that you contextualize the country in which you have performed the study, and to know if the treatments are similar in the rest of the countries.
Response 3: Data was collected from a medical center in northern Taiwan which is a member of the ELSO Registry Database Development Committee. Therefore, the standards in Taiwan for indications, techniques, and medical care of patients who undergo ECMO are similar other member countries.
Comment 4: In the discussion, the authors highlight the limitations that are quite considerable in the present study. For example, the sample size.
Response 4: The authors appreciate the reviewer’s concerns. The point we wanted to emphasize was that compared with previous studies, our sample size was considerably larger.
Comment 5: In the conclusions, the analysis of the psychological treatments that can be given to these patients is somewhat poor. It should go more deeply into this subject and say which professionals should participate in the multidisciplinary treatment mentioned. It would also be interesting to mention whether psychological help is currently provided.
Response 5: The authors appreciate the reviewer’s suggestion. We revised the Conclusion (Pages 23-24) as follows:
With the advances in medical treatments, the indicators for evaluating whether a treatment is effective should be more than the survival rate; attention should also be paid to a patient’s prognosis. This study found that 20-28.4% of survivors of ECMO had HADS-D scores suggestive of the presence of depression. The mode of ECMO was associated with depression, and the depression status of VV-ECMO patients was more severe compared with VA-ECMO. Therefore, more clinical attention should be paid to evaluating physical and mental function and social support of patients receiving VV-ECMO due to acute respiratory failure.
Thus, from the moment patients are successfully weaned from ECMO, it is recommended that continuous assessments and treatments of a patient’s mental health be conducted. A multi-professional clinical care path for patients that is established as soon as practically possible should include physicians, psychologists, neurologists, case managers, nurses, and pulmonary and cardiac physiotherapists. Prior to hospital discharge, a case manager should be assigned to the patient to provide information relative to their clinical needs for managing their health post-discharge. We suggest healthcare professionals develop a partnership with patients and provide them with information on relaxation techniques, such as breathing control, mediation or listening to their favorite music. A strong relationship between patient and healthcare professionals could facilitate initiation of self-management interventions for chronic diseases and encourage patients to establish their own physical and pulmonary rehabilitation targets. Bandura’s social cognitive theory [47] suggests self-efficacy plays an important role in determining which self-management activities a person will perform, with expectations of personal efficacy based on four major sources of information: performance accomplishments, vicarious experience, verbal persuasion, and physiological states, which can be augmented by feedback from healthcare professionals. Finally, case managers should continue to follow-up on the patient’s recovery in a timely manner and provide them with strategies for reducing psychological stress such as pharmacological or EPP therapy to help them return to a healthy lifestyle as soon as possible.
Round 2
Reviewer 1 Report
The authors have made major revision of the manuscript to address this reviewer's concerns and comments, fully integrating this reviewer’s comments into their revised manuscript, they have revised the introduction section through to the Conclusion section, particularly removing the narrative from the conclusion that is not supported by their study. Further review is left to the discretion of the authors.
Author Response
The authors have made major revision of the manuscript to address this reviewer's concerns and comments, fully integrating this reviewer’s comments into their revised manuscript, they have revised the introduction section through to the Conclusion section, particularly removing the narrative from the conclusion that is not supported by their study. Further review is left to the discretion of the authors.
Response: The authors appreciate reviewer’s decision.
Reviewer 2 Report
27 January 2021
Review on the manuscript titled “Post-Discharge Depression Status for Survivors of Extracorporeal Membrane Oxygenation (ECMO): Comparison of Veno-venous ECMO and Veno-arterial ECMO” by Lin WJ et al., submitted to the International Journal of Environmental Research and Public Health (IJERPH)
Manuscript ID: ijerph-1552660
Dear Authors,
Extracorporeal membrane oxygenation (ECMO) provides prolonged cardiac and respiratory support to sustain life. The authors conducted a cross-sectional study to assess post-discharge depression in patients who received veno-venous (VV) and veno-arterial (VA) ECMO by the Hospital Anxiety and Depression Scale-Depression (HADS-D). The results showed that the scores are significantly higher in patients who received VV-ECMO than those with VA-ECMO and that the mode of ECMO is associated with depression with a greater level of post-discharge depression for participants who received VV- ECMO. The authors concluded that more psychological care is recommended for patients who receive VV-ECMO. The authors addressed their response accordingly and the manuscript is revised mostly. However, the authors refrain from discussing the pathogenesis of depression in depth, which is the center of study in this manuscript and which makes the value of manuscript limited.
Please consider the following:
- Introduction, discussion:
I believe that more information on neurobiology and current treatment of depression would be necessary to truly provide a more thorough analysis of differences in depressive symptoms between survivors of VV- and VA-ECMO: in this regard, I suggest again to add finding from additional evidence that have focused on this topic including stress, inflammation, comorbidity (https://doi.org/10.1038/s41380-021-01326-4; https://doi.org/10.1016/j.neubiorev.2021.04.036; https://doi.org/10.1016/j.regpep.2012.08.017; https://doi.org/10.3390/biomedicines9070734; https://doi.org/10.3390/biomedicines9050517).
- References: Please cite more references, at least more than 50.
The manuscript contains two figures, three tables and 47 references, which I believe are dramatically insufficient for a research paper. The authors should include more evidence and reviews to back their claims and focus on deepening the subject of their manuscript. Finally, the manuscript carries important value presenting the difference in the occurrence of post-discharge between VA-ECMO and VV-ECMO and I recommend this manuscript for publication after major revision.
I declare no conflict of interest regarding this manuscript.
This review is contributed by Dr. Simone Battaglia, University of Bologna.
Best regards,
Masaru Tanaka, M.D., Ph.D.
Author Response
General comment: Extracorporeal membrane oxygenation (ECMO) provides prolonged cardiac and respiratory support to sustain life. The authors conducted a cross-sectional study to assess post-discharge depression in patients who received veno-venous (VV) and veno-arterial (VA) ECMO by the Hospital Anxiety and Depression Scale-Depression (HADS-D). The results showed that the scores are significantly higher in patients who received VV-ECMO than those with VA-ECMO and that the mode of ECMO is associated with depression with a greater level of post-discharge depression for participants who received VV- ECMO. The authors concluded that more psychological care is recommended for patients who receive VV-ECMO. The authors addressed their response accordingly and the manuscript is revised mostly. However, the authors refrain from discussing the pathogenesis of depression in depth, which is the center of study in this manuscript and which makes the value of manuscript limited.
General response: The authors appreciate reviewer’s concern. We have added a description of the pathogenesis of depression in Introduction (Page 4):
Psychological problems are common in patients with heart failure [9], acute respiratory distress syndrome [10] and critical illness [11]. Depression is one of most common psychological problems, which co-exists with chronic respiratory diseases [12], symptoms of dyspnea [13], acute respiratory distress syndrome (ARDS) [13,14], and disorders of anxiety and dementia [15]. Survivors of ARDS and patients with long periods of hospitalization can experience high levels of depression, anxiety, and post-traumatic stress syndrome for up to 5 years following hospital discharge [16,17], which may be associated with increased levels of mediators of inflammation [12], ventromedial prefrontal cortex activity [18], and changes in components of the tryptophan-kynurenine metabolic response to inflammation [19].
Comment 1: Introduction, discussion
I believe that more information on neurobiology and current treatment of depression would be necessary to truly provide a more thorough analysis of differences in depressive symptoms between survivors of VV- and VA-ECMO: in this regard, I suggest again to add finding from additional evidence that have focused on this topic including stress, inflammation, comorbidity (https://doi.org/10.1038/s41380-021-01326-4; https://doi.org/10.1016/j.neubiorev.2021.04.036; https://doi.org/10.1016/j.regpep.2012.08.017; https://doi.org/10.3390/biomedicines9070734; https://doi.org/10.3390/biomedicines9050517).
Response 1: We thank the reviewer for these suggestions. Our group does not include a neurologist or neuropsychologist. The possible causes for these differences would be an important topic of investigation for researchers in the fields of neuroscience or neuropsychiatry. Please see Response 1, above, which includes additional information on the neurobiology of depression now included on Page 4 of the Introduction.
We have also revised the Discussion
Pages 22-23:
Patients with ARDS and other critical illnesses face numerous stressors. Survivors of ARDS and others with long periods of hospitalization often have long recovery periods and high levels of psychological distress [17]. In a 5-year prospective longitudinal cohort study of 186 ARDS survivors, 32% of survivors were affected by depression [17]. Most participants in our study who underwent VV-ECMO were survivors with ARDS. Healthcare professionals need to identify risk factors for prolonged psychiatric morbidity in these survivors of ARDS, which have been shown to include lower socioeconomic status, prior psychiatric morbidity, and worse baseline physical functioning [17].
Pharmacotherapy, such as serotonin re-uptake inhibitors (SSRIs), in combination with psychotherapy is generally regarded as the first line of treatment for patients with anxiety and depressive disorders [53]. Although it remains unclear whether endogenous kynurenines contribute directly to the initiation of neuropathological changes, it is well-established that the dysregulation of the metabolic kynurenine pathway is involved in the pathophysiology of several inflammation-linked neuropsychiatric diseases. However, manipulation of the kynurenine pathway through the administration of enzyme inhibitors may serve as a therapeutic strategy for neuropsychiatric disease [18,51]. New evidence for the implementation of new treatments includes the effectiveness of non-invasive brain simulation (NIBS), which interferes with and modulates abnormal activity of neural circuits via the amygdala-mPFC-hippocampus pathway, which is involved in the acquisition and consolidation of fear memories that are altered in many psychiatric disorders including depression [52]. Existential phenomenological psychotherapy (EPP) has been demonstrated to augment pharmacological support for patients with mood disorders for nearly a century, which helps patients find meaning and purpose in life to override physical limitations [53]. Therefore, we suggest early evaluations of patients who have undergone ECMO to enable timely support strategies that may combine pharmacological, NIBS and psychological support.
We have cited Borgomaneri and colleagues in the last paragraph of the discussion (Page 24) as follows:
The mental health of all ECMO patients is worthy of the attention of healthcare professionals. Therefore, we strongly encourage researchers in the fields of neuroscience and neuropsychiatry to examine the neurophysiological impact of ECMO on areas of the prefrontal cortex, which has been shown to be involved in depression [54]. Second, the use NIBS techniques should be investigated as a valid alternative in the treatment of those patients that do not respond to psychotherapy and/or drug treatments [55].
Comment 2: References
Please cite more references, at least more than 50.
The manuscript contains two figures, three tables and 47 references, which I believe are dramatically insufficient for a research paper. The authors should include more evidence and reviews to back their claims and focus on deepening the subject of their manuscript. Finally, the manuscript carries important value presenting the difference in the occurrence of post-discharge between VA-ECMO and VV-ECMO and I recommend this manuscript for publication after major revision.
Response 2: We thank the reviewer for these comments. We have revised the paper to improve readability and the total number of references is now 58.
Reviewer 3 Report
Dear Authors.
You have made all the proposed changes.
Thank you
Author Response
The authors appreciate reviewer’s decision.